# Barriers and facilitators perceived by healthcare professionals for implementing lifestyle interventions in patients with osteoarthritis: a scoping review

Sjoukje E Bouma ![ORCID],[1] Juliette F E van Beek,[1,2] Ron L Diercks ![ORCID],[1] Lucas H V van der Woude ![ORCID],[2,3,4] Martin Stevens ![ORCID],[1] Inge van den Akker-Scheek ![ORCID] [1]

MS and IvdA-S contributed equally.

For numbered affiliations see end of article.

**Correspondence to**
Sjoukje E Bouma;
s.e.bouma@umcg.nl

## ABSTRACT

**Objective** To provide an overview of barriers and facilitators that healthcare professionals (HCPs) perceive regarding the implementation of lifestyle interventions (LIs) in patients with hip and/or knee osteoarthritis (OA).

**Design** Scoping review.

**Data sources** The databases PubMed, Embase, CINAHL, PsycINFO and the Cochrane Library were searched from inception up to January 2021.

**Eligibility criteria** Primary research articles with a quantitative, qualitative or mixed-methods design were eligible for inclusion if they reported: (1) perceptions of primary and/or secondary HCPs (population); (2) on implementing LIs with physical activity and/or weight management as key components (concept) and (3) on conservative management of hip and/or knee OA (context). Articles not published in English, German or Dutch were excluded.

**Data extraction and synthesis** Barriers and facilitators were extracted by two researchers independently. Subsequently, the extracted factors were linked to a framework based on the Tailored Implementation for Chronic Diseases checklist.

**Results** Thirty-six articles were included. In total, 809 factors were extracted and subdivided into nine domains. The extracted barriers were mostly related to non-optimal interdisciplinary collaboration, patients' negative attitude towards LIs, patients' low health literacy and HCPs' lack of knowledge and skills around LIs or promoting behavioural change. The extracted facilitators were mostly related to good interdisciplinary collaboration, a positive perception of HCPs' own role in implementing LIs, the content or structure of LIs and HCPs' positive attitude towards LIs.

**Conclusions** Multiple individual and environmental factors influence the implementation of LIs by HCPs in patients with hip and/or knee OA. The resulting overview of barriers and facilitators can guide future research on the implementation of LIs within OA care. To investigate whether factor frequency is related to the relevance of each domain, further research should assess the relative importance of the identified factors involving all relevant disciplines of primary and secondary HCPs.

**PROSPERO registration number** CRD42019129348.

## Strengths and limitations of this study

► To our knowledge, this is the first scoping review to classify barriers and facilitators for implementing lifestyle interventions by healthcare professionals as conservative treatment for hip and/or knee osteo-arthritis in which qualitative and quantitative data were combined.

► The study population consisted of all primary and secondary healthcare professionals involved in hip and/or knee osteoarthritis care.

► Given the broad definition of 'implementing lifestyle interventions', the identified barriers and facilitators provide insight into the full spectrum of influencing factors rather than being applicable to every single way of implementing lifestyle interventions.

► Grey literature was not included in the search and selection process.

## INTRODUCTION

Regular physical activity and weight management are recommended by national and international clinical guidelines for the conservative management of hip and/or knee osteoarthritis (OA).[1–5] Previous studies have demonstrated that lifestyle interventions (LIs) focusing on exercise, alone or combined with dietary weight loss, are able to reduce hip and/or knee OA-related disability and to postpone or even prevent total joint arthroplasty.[6–10] However, these positive results are not always transferred from research settings to daily practice, which means that LIs are underused.[11] This suboptimal implementation of LIs as treatment for hip and/or knee OA can result from factors related to the patient, the healthcare professional (HCP) or the societal context.[12] Research on adhering to LIs has so far focused mainly on identifying barriers and facilitators at the patient level. However, these studies have also shown that HCPs can have a facilitating role

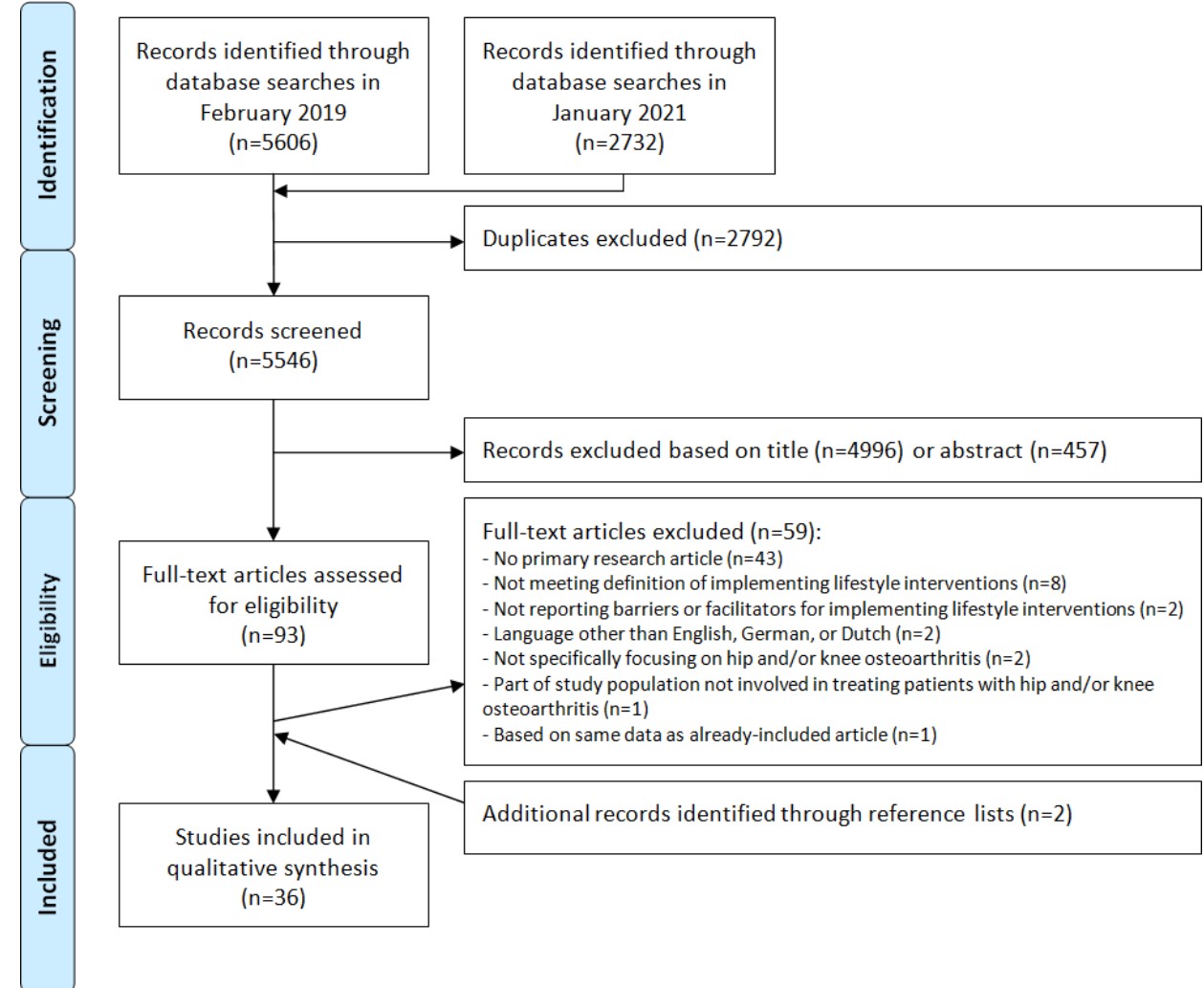

**Figure 1** Flow chart of the study selection process.

in the lifestyle behaviour of their patients, for example by providing advice, education, encouragement and instructions.[13 14]

Some research has already been conducted investigating the perspective of HCPs and the implementation of LIs in their daily practice. This knowledge is needed in order to enhance the implementation of LIs. As far as the authors know, no (systematic) literature review has previously been performed that identified and/or classified barriers and facilitators for implementing LIs in the conservative treatment of hip and/or knee OA from the perspective of all HCPs involved. One systematic review focused on the views towards OA management based on recommendations in clinical practice guidelines of HCPs working in primary care.[15] However, HCPs working in secondary care are also involved in the treatment of patients with OA, which draws attention to the importance of collaboration and communication between primary and secondary care practitioners.[16] Therefore, a scoping review was conducted aiming to provide a comprehensive overview of barriers and facilitators perceived by primary and secondary HCPs regarding the implementation of LIs in patients with hip and/or knee OA. The Tailored Implementation for Chronic Diseases

(TICD) checklist was used to guide data synthesis.[17] Within the context of this review, implementation was defined as the use of LIs as conservative treatment for hip and/or knee OA by individual HCPs.

## METHOD
### Study design
A scoping review has been defined as follows by Colquhoun *et al*: 'a form of knowledge synthesis that addresses an exploratory research question aimed at mapping key concepts, types of evidence and gaps in research related to a defined area or field by systematically searching, selecting and synthesising existing knowledge'.[18] Therefore, a scoping review was considered a suitable methodology to summarise existing literature on barriers and facilitators for implementing LIs in hip and/or knee OA and to identify potential gaps in the current literature on participation of primary and secondary HCPs. We conducted this scoping review according to the framework developed by Arksey and O'Malley.[19] Five stages were followed successively: (1) identifying the research question; (2) identifying relevant studies; (3) study selection; (4) charting the data and (5) collating, summarising and

**Table 1** Overview of included studies

| Reference | Country and health setting | Study focus | Type of data extracted | Data collection method | Data analysis method | Participants |
|---|---|---|---|---|---|---|
| Allison (2019)[27] | Australia (private primary care and public hospital care or community health) | Attitudes and perceptions towards role in weight management (knee OA) | Qualitative | Individual interviews | Inductive thematic analysis | PT (n=13, 61% female, age range 27–61 years) |
| Bossen (2016)[28] | The Netherlands (private practice) | Development and feasibility of the blended exercise therapy intervention 'e-Exercise' (hip and/or knee OA) | Qualitative | 1. Focus group 2. Individual interviews | 1. Summarising 2. Thematic trend analysis | 1. PT (n=7) 2. PT (n=5) |
| Christiansen (2020)[29] | Canada (academic and community family health practice) | Experiences with and barriers to prescribing exercise (knee OA) | Qualitative | Individual interviews | Constant comparison approach | Physician (n=11) |
| Davis (2018)[30] | Canada (single assessment centre) | Implementation of the 'GLA:D Canada' programme (hip and/or knee OA) | Qualitative | Individual interviews | Thematic content analysis | PT (n=3) |
| de Rooij (2014)[31] | The Netherlands (rehabilitation centre) | Development of comorbidity-adapted exercise protocols (knee OA) | Qualitative | Individual interviews | Analysing notes | PT (n=3) |
| Egerton (2017)*[32] | Australia (primary care) | Perspectives on potential barriers and facilitators to engagement with a proposed model of service delivery for primary care management (knee OA) | Qualitative | Individual interviews | Interpretive thematic analysis | GP (n=11, 64% female, mean age 50.8 years (range: 34–67)) |
| Egerton (2018)*[33] | Australia (primary care) | Barriers and facilitators influencing clinical practice guideline implementation in primary care (knee OA) | Qualitative | Individual interviews | Interpretive thematic analysis | GP (n=11, 64% female) |
| Hinman (2016)[34] | Australia (private practice) | Experiences of being involved in delivering an integrated programme of PT-supervised exercise and telephone coaching (knee OA) | Qualitative | Individual interviews | Thematic analysis informed by grounded theory | PT (n=10, 50% female, mean age 43 years (SD: 13)) Telephone coach (n=4; 100% female, mean age 42 years (SD: 11)) |
| Hinman (2017)[35] | Australia (not specified) | Experiences using Skype as a service delivery model for PT-prescribed exercise management (knee OA) | Qualitative | Individual interviews | Thematic and constant comparative analytical approach | PT (n=8, 50% female, mean age 39 years (SD: 9)) |
| Knoop (2020)[36] | The Netherlands (primary care) | Feasibility of a newly developed model of stratified exercise therapy in primary care (knee OA) | Qualitative | 1. Individual interviews 2. Focus group | Analysed descriptively | 1. PT (n=9) 2. PT (n=14) |
| Law (2019)[37] | UK (leisure centre) | Experiences and views of referring and delivering professionals regarding the 'Lifestyle Management Programme' (hip and/or knee OA) | Qualitative | 1. Focus groups 2. Individual interviews | Framework analysis method | 1. Dietician (n=2) Exercise professional (n=3) PT (n=4) Triaging clinician (n=1) 2. GP (n=3) Total group: 46% female |
| Lawford (2019)[38] | Australia (private and public practice) | Preintervention and postintervention perceptions of telephone-delivered exercise therapy (knee OA) | Qualitative | Individual interviews | Thematic analysis approach | PT (n=8, 50% female) |
| Lawford (2020)[39] | Australia (private and public practice) | Experiences and perceptions with prescribing a strengthening exercise programme for people with comorbid obesity (knee OA) | Qualitative | Individual interviews | Inductive thematic approach | PT (n=7, 14% female) |
| Lawford (2021)[40] | Australia (private and public practice) | Experiences with a multicomponent dietary weight loss programme (knee OA) | Qualitative | Individual interviews | Thematic approach informed by grounded theory | Dietician (n=5, 100% female) |

Continued

**Table 1** Continued

| Reference | Country and health setting | Study focus | Type of data extracted | Data collection method | Data analysis method | Participants |
|---|---|---|---|---|---|---|
| MacKay (2018)*[41] | Canada (community-based and outpatient setting) | Factors influencing physical therapy management (knee OA) | Qualitative | Individual interviews | Inductive thematic analysis | PT (n=33, 76% female) |
| MacKay (2020)*[42] | Canada (community-based and outpatient setting) | Perceptions related to physical therapy management (knee OA) | Qualitative | Individual interviews | Inductive thematic analysis | PT (n=33, 76% female) |
| Mann (2011)[43] | UK (primary and secondary care) | Perceptions of current service provision and possible service improvements (hip and/or knee OA) | Qualitative | Individual interviews | Framework method | GP (n=2) Nurse practitioner (n=1) Occupational therapist (n=1) OS (n=2) Practice nurse (n=3) PT (n=2) RH (n=1) |
| Miller (2020)[44] | USA (large academic medical centre) | Barriers and facilitators to guideline-based treatment (hip and/or knee OA) | Qualitative | Individual interviews | Conventional content analysis | Physician (n=6, 50% female) |
| Nielsen (2014)[45] | Australia (not specified) | Perspectives on and experiences with an intervention of exercise combined with cognitive behavioural therapy (Pain Coping Skills Training) and the implementation process (knee OA) | Qualitative | Individual interviews | Framework analysis | PT (n=8, 88% female, age range 35–58 years) |
| Okwera (2019)[46] | UK (general practice within NHS) | Beliefs on physiotherapy management in primary care (hip and/or knee OA) | Qualitative | Individual interviews | Framework analysis | GP (n=8, 50% female, age range 31–60 years) |
| Poitras (2010)[47] | France (general practice; work setting PTs not specified) | Barriers to use of conservative management recommendations (knee OA) | Qualitative | Focus groups | Thematic content analysis | GP (n=7, 29% female, median age 53 years (range: 48–77)) PT (n=10, 40% female, median age 46.5 years (range: 24–69)) |
| Rosemann (2006)[48] | Germany (general practice) | Problems and needs for improving primary care (hip and/or knee OA) | Qualitative | Individual interviews | Description of coding process, but no specific method reported | GP (n=20, 20% female, mean age 43.5 years (range: 33–57)) Practice nurse (n=20, 100% female, mean age 41.3 years (range: 29–56)) |
| Selten (2017)[49] | The Netherlands (general practice; work setting PTs, OSs and RHs not specified) | Views on non-pharmacological, non-surgical management (hip and/or knee OA) | Qualitative | Individual interviews | Thematic analysis | GP (n=5) OS (n=7) PT (n=7) RH (n=5) Total group: 50% female, age range 24–64 years |
| Tang (2020)[50] | Australia (large metropolitan public health service) | Application of clinical practice guidelines (knee OA) | Qualitative | Individual interviews | Thematic analysis | PT (n=18) |
| Teo (2020)[51] | Australia (private practice and tertiary or non-tertiary hospitals) | Experiences with delivering care (knee OA) | Qualitative | Individual interviews | Inductive thematic approach | PT (n=22, 50% female, mean age 34 years (SD: 8, range: 24–54)) |
| Wallis (2020)[52] | Australia (general practice; OSs and RHs working in private and public hospitals) | Perceptions about management including barriers and enablers for referral to the 'GLA:D Australia' programme (hip and/or knee OA) | Qualitative | Individual interviews | Inductive thematic analysis | GP (n=5) OS (n=6) RH (n=4) Total group: mean age 52 years (SD: 12) |
| Cottrell (2016)[53] | UK (general practice) | Attitudes and beliefs regarding exercise (knee OA) | Quantitative | Survey (RR: 17%) | Descriptive statistics (frequency) | GP (n=835, 51% female) |

Continued

**Table 1**    Continued

| Reference | Country and health setting | Study focus | Type of data extracted | Data collection method | Data analysis method | Participants |
|---|---|---|---|---|---|---|
| Duarte (2019)[54] | Portugal (not specified) | Development and acceptability of the Portuguese version of the 'Fit & Strong!' programme (hip and/or knee OA) | Quantitative | Survey (RR: 100%) | Not reported | Programme instructor (n=2) |
| Hill (2018)[55] | UK (specialist practice in knee surgery) | Opinions and practices regarding the management of symptomatic OA in obesity (knee OA) | Quantitative | Survey (RR: 52%) | Descriptive statistics (frequency) | OS (n=205) |
| Hill (2018)[56] | UK (general practice) | Opinions and practices regarding the management of symptomatic OA in obesity (knee OA) | Quantitative | Survey (RR: 75%) | Descriptive statistics (frequency) | GP (n=130) |
| Hofstede (2016)[57] | The Netherlands (52% of OSs worked at a general hospital) | Barriers and facilitators associated with prescription of different non-surgical treatments (hip and/or knee OA) | Quantitative | Survey (RR: 36%) | Descriptive statistics (frequency) | OS (n=172, 9% female, mean age 48.4 years (SD: 8.6)) |
| Lawford (2018)[58] | Australia (private and public practice) | Perceptions of remotely delivered service models for exercise management (hip and/or knee OA) | Quantitative | Survey (RR: unknown) | Descriptive statistics (frequency and level of agreement) | PT (n=217, 72% female) |
| Reid (2014)[59] | New Zealand (general practice; work setting OSs not specified) | Self-reported behaviour, experiences, expectations and perceptions regarding physiotherapy referral and management (hip and/or knee OA) | Quantitative | Survey (RR: 46% (GP) and 26% (OS)) | Descriptive statistics (frequency) | GP (n=24) OS (n=20) Total group: 34% female, mean age 52.2 years (SD: 8.5) |
| de Rooij (2020)[60] | The Netherlands (primary care) | Facilitators and barriers for usage of a strategy for exercise prescription in patients with comorbidity (knee OA) | Mixed-methods | 1. Survey (RR: 100%) 2. Individual interviews | 1. Descriptive statistics (frequency) 2. Summarising notes | 1. PT (n=34, 68% female, mean age 43.7 years (SD: 11.1)) 2. PT (n=10) |
| Holden (2009)[61] | UK (NHS and non-NHS) | Attitudes and beliefs regarding exercise (knee OA) | Mixed-methods | 1. Survey (RR: 58%) 2. Individual interviews | 1. Descriptive statistics (level of agreement) 2. Thematic analysis | 1. PT (n=538, 87% female) 2. PT (n=24, 67% female) |
| Kloek (2020)[62] | The Netherlands (primary care practice) | Experiences with and determinants related to the usage of the blended physiotherapy intervention 'e-Exercise' (hip and/or knee OA) | Mixed-methods | 1. Survey (RR: 40%) 2. Individual interviews | 1. Descriptive statistics (frequency) 2. Grounded theory methodology | 1. PT (n=49) 2. PT (n=9, 33% female, median age 52 years (range: 24–59)) |

*Data for both studies were collected during the same interview.
GLA:D, Good Life with osteoArthritis in Denmark; GP, general practitioner; NHS, National Health Service; OA, osteoarthritis; OS, orthopaedic surgeon; PT, physiotherapist; RH, rheumatologist; RR, response rate.

reporting the results.[19] The Preferred Reporting Items for Systematic Reviews and Meta-Analyses Extension for Scoping Reviews checklist was used as reporting guideline.[20]

### Data sources and searches

A search strategy was developed consisting of four components: search terms related to: (1) primary and secondary HCPs; (2) hip and/or knee OA; (3) LIs and (4) barriers and facilitators. This search strategy was applied in five bibliographic electronic databases (ie, PubMed, Embase, CINAHL, PsycINFO and the Cochrane Library) to identify relevant articles from inception up to 19 January 2021. A detailed search strategy for each of the databases can be found in online supplemental file 1. Reference lists of included articles were manually searched for additional relevant articles. Primary research articles with a quantitative, qualitative or mixed-methods design were eligible for

inclusion; study protocols, reviews, abstracts and commentaries were excluded. Articles written in English, German or Dutch were eligible for inclusion. No restrictions were applied regarding publication period.

### Study selection

Eligibility criteria were described according to the population–concept–context framework.[21] First, the study population was defined as all primary and secondary HCPs who are involved in the conservative treatment of patients with hip and/or knee OA. This definition includes, respectively, HCPs providing general medical care and HCPs providing more specialised care (with or without a referral). Articles focusing solely on the perspective of patients with hip and/or knee OA were excluded. Second, the concepts central to this review were barriers and facilitators for implementing LIs. Barriers and facilitators were defined

**Table 2** Distribution of the extracted factors per included article across the domains, which were largely based on the Tailored Implementation for Chronic Diseases checklist

| Reference | Domain 1: Intervention factors | Domain 2: Individual HCP factors | Domain 3: Patient factors | Domain 4: Professional interactions | Domain 5: Incentives and resources | Domain 6: Capacity for organisational change | Domain 7: Social, political and legal factors | Domain 8: Patient and HCP interactions | Domain 9: Disease factors | Total no of factors in article |
|---|---|---|---|---|---|---|---|---|---|---|
| Allison (2019)[27] | | 3 | | 2 | 2 | 1 | | 4 | | 12 |
| Bossen (2016)[28] | 8 | | | | | | | | | 8 |
| Christiansen (2020)[29] | 1 | 5 | 2 | | | | | | 1 | 9 |
| Davis (2018)[30] | 6 | | 1 | | | | | | | 7 |
| De Rooij (2014)[31] | 3 | 2 | | | | | | | | 5 |
| Egerton (2017)[32] | 20 | 3 | 1 | 9 | 3 | | | | 1 | 37 |
| Egerton (2018)[33] | 5 | 9 | 5 | | 6 | | 1 | 1 | 5 | 32 |
| Hinman (2016)[34] | 7 | 1 | 2 | 10 | | | | | | 20 |
| Hinman (2017)[35] | 18 | | | | | | | | | 18 |
| Knoop (2020)[36] | 4 | | 1 | 1 | | | | | | 6 |
| Law (2019)[37] | 8 | 1 | 5 | 1 | 2 | | | 1 | | 18 |
| Lawford (2019)[38] | 26 | | | | | | | | | 26 |
| Lawford (2020)[39] | 11 | | 7 | | | | | 1 | | 19 |
| Lawford (2021)[40] | 12 | | 3 | | | | | | | 15 |
| MacKay (2018)[41] | 6 | 5 | 14 | 7 | 6 | 2 | 1 | | | 41 |
| MacKay (2020)[42] | 4 | 12 | 5 | 1 | 1 | | | 4 | | 27 |
| Mann (2011)[43] | 2 | 1 | 4 | 10 | 1 | | | | 1 | 19 |
| Miller (2020)[44] | 4 | 4 | 7 | 3 | 8 | | 1 | 1 | 1 | 29 |
| Nielsen (2014)[45] | 13 | 8 | 1 | | 3 | 2 | | | | 27 |
| Okwera (2019)[46] | 4 | 6 | 6 | 12 | | | 2 | | 2 | 32 |
| Poitras (2010)[47] | 11 | 13 | 19 | 3 | | | | 1 | 5 | 52 |
| Rosemann (2006)[48] | 1 | 4 | 5 | 4 | 6 | | 1 | 1 | 1 | 23 |
| Selten (2017)[49] | 7 | 3 | 3 | 14 | 2 | | | 4 | | 33 |
| Tang (2020)[50] | | 12 | 4 | | | | | 1 | | 17 |
| Teo (2020)[51] | 3 | 11 | 8 | | | | | | 1 | 23 |
| Wallis (2020)[52] | 17 | | 7 | 3 | 2 | | | | 1 | 30 |
| Cottrell (2016)[53] | 12 | 10 | 4 | 2 | 3 | | | | | 31 |
| Duarte (2019)[54] | 1 | | 2 | | | | | | | 3 |

Continued

**Table 2** Continued

| Reference | Domain 1: Intervention factors | Domain 2: Individual HCP factors | Domain 3: Patient factors | Domain 4: Professional interactions | Domain 5: Incentives and resources | Domain 6: Capacity for organisational change | Domain 7: Social, political, and legal factors | Domain 8: Patient and HCP interactions | Domain 9: Disease factors | Total no of factors in article |
|---|---|---|---|---|---|---|---|---|---|---|
| Hill (2018)[55] | | 5 | | 2 | | | | | | 7 |
| Hill (2018)[56] | 2 | 4 | | 2 | | | | | | 8 |
| Hofstede (2016)[57] | 5 | 3 | | 4 | 1 | 1 | | | | 14 |
| Lawford (2018)[58] | 33 | | | | | | | | | 33 |
| Reid (2014)[59] | 4 | 1 | 3 | 1 | | | | | | 9 |
| De Rooij (2020)[60] | 18 | 8 | 4 | 9 | 2 | 1 | 3 | | | 45 |
| Holden (2009)[61] | 13 | 10 | 14 | | 3 | | | | 2 | 42 |
| Kloek (2020)[62] | 26 | | | 1 | 5 | | | | | 32 |
| Total no of factors in domain | 315 | 144 | 137 | 101 | 56 | 7 | 9 | 19 | 21 | 809 |

HCP, healthcare professional.

as any belief, experience, factor, opinion, reason or view reported by an HCP that potentially influences (either impedes or facilitates) implementation of LIs in patients with hip and/or knee OA. These barriers and facilitators were extracted from both quantitative (eg, survey) and qualitative (eg, interview) data. Implementing LIs was broadly defined, ranging from mentioning or discussing a healthy lifestyle to recommending or running specific lifestyle programmes, as long as it was clearly described that physical activity and/or weight management were key components. This definition includes physiotherapeutic exercise interventions (aerobic, functional or strengthening programmes), dietary interventions and self-management programmes. Physiotherapeutic modalities such as acupuncture, manual therapy, and massage, and self-management programmes whose content was not specified were not considered LIs and were therefore excluded. Physical activity was also broadly defined, ranging from physical activity during activities of daily living to participation in supervised or non-supervised exercise therapy or sports. Articles not primarily focusing on implementing LIs (eg, development and evaluation of clinical guidelines, general management of hip and/or knee OA, general patient–practitioner relationship or shared decision making) also fell outside the scope of this review. Lastly, the context of this review was the conservative treatment of hip and/or knee OA in both primary and secondary healthcare settings. Articles focusing on preoperative or postoperative treatment of hip and/or knee OA were excluded. Two researchers (SB together with AJ or JvB) independently assessed the eligibility of the identified articles based on the above criteria in three consecutive rounds: (1) based on title; (2) abstract and (3) full text of the article. Any disagreements among the researchers were resolved in consensus meetings.

### Data extraction and quality assessment

A data extraction form was created and pilot-tested in order to systematically record study characteristics (first author, year of publication, country of origin, aims/purpose, study design, data collection method, data analysis method, theoretical basis, study population, setting, recruitment method, type of LI, patient population) and outcomes (barriers, facilitators and/or unclear factors (ie, an influencing factor, but not clearly defined as barrier or facilitator)). Study quality was assessed with the Mixed Methods Appraisal Tool (MMAT). The MMAT is a critical appraisal tool that can be used in reviews of mixed studies to assess the methodological quality of different study design categories: mixed-methods, qualitative and quantitative studies (randomised controlled trials, non-randomised studies and descriptive studies).[22 23] Since calculating a total score is discouraged,[23] it was chosen to present the ratings of the individual criteria.

Data extraction was performed in two stages. The first stage consisted of filling in the data extraction form and the MMAT for each article, done by two researchers

**Table 3** Overview of barriers, facilitators and unclear factors that influence the implementation of LIs as perceived by HCPs for all domains, which were largely based on the Tailored Implementation for Chronic Diseases checklist*

| Category | Subcategory—barriers | Subcategory—facilitators | Subcategory—unclear factors |
|---|---|---|---|
| *Domain 1: Intervention factors (factors related to LIs)* | | | |
| Effectiveness | ▶ LIs have little or no effect on OA[29 32 33 44 46 47 49 53 59 61] <br> ▶ Potential effects of LIs are difficult to accomplish.[47 48 53 61] | ▶ LIs have positive effects on affected joint(s).[35 38 40–42 47 49 52 53 58 61] <br> ▶ LIs have positive effects on general health.[33 40 47 49 56 57] <br> ▶ LIs have positive mental effects.[30 35 37 38 40 49 52 57] <br> ▶ LIs have positive effects (not further specified).[34 37 44 49 52 54 57] | |
| Safety | ▶ LIs are unsafe or have negative effects.[39 47 52 61] | ▶ LIs are safe.[53 57] <br> ▶ Research environment or protocols provide a safety net.[31 35 38 39] | |
| Design | ▶ Non-optimal content or structure of LIs.[34 36 52 53 62] <br> ▶ Challenges for patients during participation in LIs.[39 40 45] <br> ▶ Challenges for HCPs during delivery of LIs.[28 30 39 60 62] | ▶ Positive experiences with or suggestions to improve the content or structure of LIs.[28 30 34 37 40 45 52 60 62] <br> ▶ Ease for patients during participation in LIs.[39 40 52] <br> ▶ Ease for HCPs during delivery of LIs.[30 31 34 39 45 60 62] | |
| Personalised treatment | ▶ Insufficient ability to provide personalised treatment within LIs.[32 45 62] | ▶ Ability and importance of providing personalised treatment within LIs.[37 39 42 45 47 51 53 60–62] | |
| Accessibility | ▶ LIs are unavailable or inaccessible.[28 33 41 43 44 53 56 59 61] <br> ▶ Costs of LIs to patients.[32 33 41 44 51 52] <br> ▶ LIs are not feasible or sustainable.[32 60] <br> ▶ Inconvenience to patients when accessing LIs.[51–53] | ▶ LIs are available or accessible, or suggestions for improvement.[32 37 41 46 57 59] <br> ▶ LIs are feasible or sustainable.[32 36 37 42 60] <br> ▶ Convenience for patients when accessing LIs.[52] | |
| Telehealth | ▶ Disadvantages of telehealth in terms of effectiveness[32 58 62] <br> ▶ Telehealth is not safe for patients or patient/data privacy.[32 58] <br> ▶ Challenges for HCPs regarding lack of physical/visual contact.[35 38 58 62] <br> ▶ Other challenges for HCPs regarding feasibility of telehealth.[28 32 35 38 58 62] <br> ▶ Patient-related challenges regarding feasibility of telehealth.[28 32 62] <br> ▶ Negative aspects regarding communication and relationship using telehealth.[34 35 38 40] | ▶ Benefits of telehealth in terms of effectiveness.[28 35 38 58 62] <br> ▶ Telehealth is safe for patients or patient/data privacy.[35 58 62] <br> ▶ Lack of physical/visual contact not a major issue for HCPs.[35 38 58] <br> ▶ Positive attitude or needs of HCPs regarding feasibility of telehealth.[35 38 40 58 62] <br> ▶ Patient-related benefits regarding feasibility of telehealth.[28 32 35 38 58] <br> ▶ Positive aspects regarding communication and relationship using telehealth.[38 40] | |
| *Domain 2: Individual HCP factors (factors related to individual primary and secondary HCPs)* | | | |
| Expertise | ▶ Lack of knowledge or skills around LIs or promoting behavioural change.[27 29 33 41 42 45 47 49–51 56 60 61] <br> ▶ Lack of knowledge or skills around OA care in general.[43 44 46 48] <br> ▶ Lack of knowledge or skills around specific resources.[33 50 60] | ▶ Having or improving knowledge or skills around LIs or promoting behavioural change.[33 34 41 42 45 46 50] <br> ▶ Having or improving knowledge or skills around OA care in general.[33 44 46 48] <br> ▶ Available resources might improve knowledge and decision-making.[31 50 60] | ▶ Clinical experience[42] |
| Attitude | ▶ Negative attitude towards LIs.[29 53 61] <br> ▶ Negative attitude towards guidelines or protocols.[46] | ▶ Positive attitude towards LIs.[33 41 42 45–47 50 51 53 55–57 59] <br> ▶ Positive attitude towards guidelines or protocols.[27 57 60] | ▶ Autonomy[37] |
| Role | ▶ Perception of own role potentially impeding prescription or follow-up of LIs.[29 33 42 44 47–51 53 55 61] <br> ▶ Negative consequences for own role when referring patients to LIs.[32] | ▶ Perception of own role potentially stimulating prescription or follow-up of LIs.[33 41 42 47 48 51 53 55 56 61] <br> ▶ Positive consequences for own role when referring patients to LIs.[32] | |
| *Domain 3: Patient factors (factors related to patients with hip and/or knee OA)* | | | |
| Health status | ▶ Severity of disease and symptoms[32 44 47 50 52 61] <br> ▶ Negative impact of comorbidities.[29 39 44 47 48 51 52] <br> ▶ Other patient characteristics.[47 52 59] | ▶ Severity of disease and symptoms.[39 47 53 59 61] <br> ▶ Other patient characteristics.[41 51 59] | ▶ Severity of disease and symptoms.[42 46 53 61] <br> ▶ Other patient characteristics.[41] |
| Treatment expectations and preferences | ▶ Negative attitude towards LIs[29 33 34 36 39 41–48 51–53 60 61] <br> ▶ Positive attitude towards TJA[37 43 48] | ▶ Make use of patients' preference for TJA within LIs[37] | ▶ Patients' preferences[46] |
| Active participation | ▶ Low patient adherence or engagement[33 37 41 42 46 47 51 54 61] | ▶ High patient adherence or engagement[34 39 40 54] <br> ▶ Importance of high patient adherence or engagement for effectiveness of LIs[30 41 42 47 53 61] | |

Continued

**Table 3** Continued

| Category | Subcategory—barriers | Subcategory—facilitators | Subcategory—unclear factors |
|---|---|---|---|
| Capabilities | ▶ Low health literacy[33 37 39–41 43 44 47 49 51 52 60 61]<br>▶ Limited financial resources[41 44]<br>▶ Other responsibilities[41 52] | ▶ High health literacy or importance of education[39 42 43 49 51 60]<br>▶ Social support[40 48] | ▶ Health literacy[46]<br>▶ Other responsibilities[41] |
| *Domain 4: Professional interactions (factors related to interactions between primary and secondary HCPs)* | | | |
| Collaboration | ▶ Non-optimal interdisciplinary collaboration or healthcare provision[27 32 34 41 43 46 47 49 53 60]<br>▶ No access to other HCPs[41] | ▶ Good interdisciplinary collaboration or healthcare provision, or suggestions for improvement[27 32 34 37 41 43 44 46–49 52 53 55–57 59 60 62]<br>▶ Access to other HCPs[32 41–43 46] | |
| Communication and referral | ▶ Lack of communication between HCPs[46 48 60]<br>▶ Challenges of communication and referral procedures[34 36 44 46 60] | ▶ Improving communication between HCPs[32 34 46 48 52 57]<br>▶ Needs regarding communication and referral procedures[32 41 46 49 52] | |
| *Domain 5: Incentives and resources (factors related to the availability of incentives and resources for primary and secondary HCPs)* | | | |
| Time | ▶ Lack of time within patient consultations[33 41 43–45 49 53 61]<br>▶ Lack of time due to other demands (or not further specified)[32 37 41 48 62] | ▶ Adequate duration of patient consultations[33 41]<br>▶ Adequate duration of specific interventions or protocols[32 45 60 62] | |
| Financial resources | ▶ Limited financial resources within organisation[45 48] | ▶ Financial reward for implementing LIs[32 48 60] | |
| Information resources | ▶ Lack of information resources[27 37 44 48]<br>▶ Challenges in accessing information resources[41 44 53] | ▶ Availability of information resources[27 44 52 57]<br>▶ Access to information resources[33 41 42 52] | |
| Facilities | ▶ Negative attitude towards information technology[33] | ▶ Potential use of information technology[33 44]<br>▶ Benefits of working in health centres[49] | |
| *Domain 6: Capacity for organisational change (factors related to the organisation where primary and secondary HCPs work)* | | | |
| Professional paradigm | | ▶ Adequate professional paradigm or suggestions for expansion[27 41 45] | |
| Monitoring | | ▶ Audit[57] | |
| Support within the organisation | ▶ Management not supportive[60] | | |
| *Domain 7: Social, political, and legal factors (factors related to the social, political and legal context)* | | | |
| Healthcare system | ▶ Restrictions due to health insurance[41 48 60] | ▶ Benefits of good health insurance[44 46 60]<br>▶ Government subsidies[33] | |
| *Domain 8: Patient and HCP interactions (factors related to interactions between patients with hip and/or knee OA and primary and secondary HCPs)* | | | |
| Therapeutic alliance | ▶ Potential negative influence of implementing LIs to relationship[37] | ▶ Importance of communication and relationship[39 42 48 49] | |
| Lifestyle as conversation topic | ▶ Challenges of discussing weight[27 33 42 49 50] | ▶ Factors that could ease the way to discussing weight[27 42 44 47 49] | |
| *Domain 9: Disease factors (factors related to OA)* | | | |
| Image | ▶ OA seen as low priority[29 32 43 46–48]<br>▶ OA seen as untreatable and local condition (wear-and-tear)[33 44 46 47 51 52 61] | ▶ Optimistic views towards OA[33 47] | |

HCP, healthcare professional; LI, lifestyle intervention; OA, osteoarthritis; TJA, total joint arthroplasty.

(SB/JvB) independently. Regarding barriers and facilitators, both researchers extracted the relevant units of text and/or descriptive statistics from the Results sections. Any discrepancies between the researchers in this first stage were resolved in consensus meetings. During the second stage, the extraction of barriers and facilitators was discussed among the research team (SB/MS/IvdA-S) and the process was further refined for both quantitative and qualitative data. Regarding quantitative data, factors were only extracted if ≥50% of participants indicated that the factor influenced the implementation of LIs.[24 25] For close-ended questions or attitude statements with multiple answer options, participants were classified as being 'in agreement' or 'not in agreement'. If this classification had not yet

been made by the authors of the original article, it was made based on the possible answer options, with '(strongly) agree', 'to a reasonable/large extent' and 'yes' indicating agreement, and 'neither disagree or agree', 'don't know', 'neutral', 'a little bit/not at all', '(strongly) disagree', and 'no' indicating not in agreement. Next, the factor was classified as barrier or facilitator depending on the formulation of the question and which of the two groups ('in agreement' vs 'not in agreement') comprised ≥50% of the participants. In case of open-ended questions, all mentioned factors were extracted. Regarding qualitative data, if the authors of the original study did not explicitly identify a factor as barrier or facilitator, the description in the text or the participants' quotes were used to classify the factor

as barrier (ie, impeding/negative/problem/lack), facilitator (ie, facilitating/positive/solution/need) or unclear (ie, insufficient information). In addition, all unclear factors were rediscussed with a third researcher (IvdA-S) to assess whether these factors could nevertheless be classified as barrier or facilitator. At the end of the second stage, final data extraction based on the above criteria was performed by one researcher (SB), who also checked the consistency of the entire data extraction process.

### Data synthesis and analysis

A narrative synthesis of the data was undertaken, based on the TICD checklist developed by Flottorp *et al.*[17] This checklist aims to assist in identifying key determinants of professional practice, defined as factors that might prevent or enable healthcare improvements, and is intended for use in research on implementation and quality improvement in healthcare. It consists of seven domains: (1) guideline factors; (2) individual health professional factors; (3) patient factors; (4) professional interactions; (5) incentives and resources; (6) capacity for organisational change; and (7) social, political and legal factors. The authors of the current study have previously used the TICD checklist in the analysis of focus group data on the same topic, revealing two additional domains: (8) patient and HCP interactions; and (9) disease factors.[26] One researcher (SB) assigned all extracted factors to one of these nine domains and then inductively developed different categories and subcategories of factors per domain. The resulting classification of factors and corresponding conclusions were subsequently discussed among the research team (SB/MS/IvdA-S).

### Patient and public involvement

Patients or the public were not involved in this study as the study aim did not concern patients but HCPs.

## RESULTS

### Study selection

A flow chart of the study selection process is presented in figure 1. A total of 8338 articles were retrieved. After removal of duplicates and exclusion of articles based on title or abstract, 93 potentially relevant articles remained for full-text screening. Ultimately, 36 articles were included in the qualitative synthesis.[27–62]

### Study characteristics

General characteristics of the included studies are presented in table 1. The majority of studies were conducted in Australia (36%), the Netherlands (19%), the UK (19%) and Canada (11%). Qualitative data were extracted in 26 studies (72%), quantitative data in 7 studies (19%), and both qualitative and quantitative data in the remaining 3 studies (8%). Individual interviews were most commonly used as qualitative

data collection method, while the quantitative studies were all based on cross-sectional surveys. Most studies included physiotherapists or general practitioners (or physicians) as study population. Other participants were dieticians, exercise professionals, a nurse practitioner, an occupational therapist, orthopaedic surgeons, practice nurses, programme instructors, rheumatologists, telephone coaches and triaging clinicians.

### Quality assessment

Findings of the quality assessment of the included studies based on the MMAT are shown in online supplemental file 2. Regarding the qualitative data assessments, only one study had the maximum of five positive ratings. Seven studies had a negative rating for the item on substantiating the interpretation of results, as no or a limited number of participant quotes were presented. In addition, many unknown ratings were given due to a lack of information about the applied qualitative approach and/or data analysis methods and their rationale. Regarding the quantitative data assessments, most studies had a negative or unknown rating for the risk of non-response bias due to low response rates or a lack of information about the response rate and/or reasons for non-response. In addition, the item on representativeness of the sample was often given an unknown rating because insufficient information about the sample and/or non-responders was presented. Finally, all three mixed-methods studies had a negative rating since the qualitative and quantitative components did not adhere to their specific quality criteria. For the other four mixed-methods criteria, only one of these three studies obtained positive ratings.

### Synthesis of results

A total of 809 factors were extracted from the 36 included articles. Table 2 presents the distribution of factors from the individual studies across the aforementioned nine domains, which were largely based on the TICD checklist. The highest number of factors was assigned to intervention factors (n=315), followed by individual HCP factors (n=144), and patient factors (n=137). The lowest number of factors was assigned to capacity for organisational change (n=7), followed by social, political and legal factors (n=9), and patient and HCP interactions (n=19). In table 3, the content of the nine domains is further explained by presenting an overview of the created categories and subcategories of factors that potentially influence the implementation of LIs by HCPs within each domain. A full overview of all extracted factors can be found in online supplemental file 3 (presented per domain) and online supplemental file 4 (presented per article).

### Categories

The distribution of barriers and facilitators across the various categories is presented in figure 2. The highest number of barriers was assigned to the following five

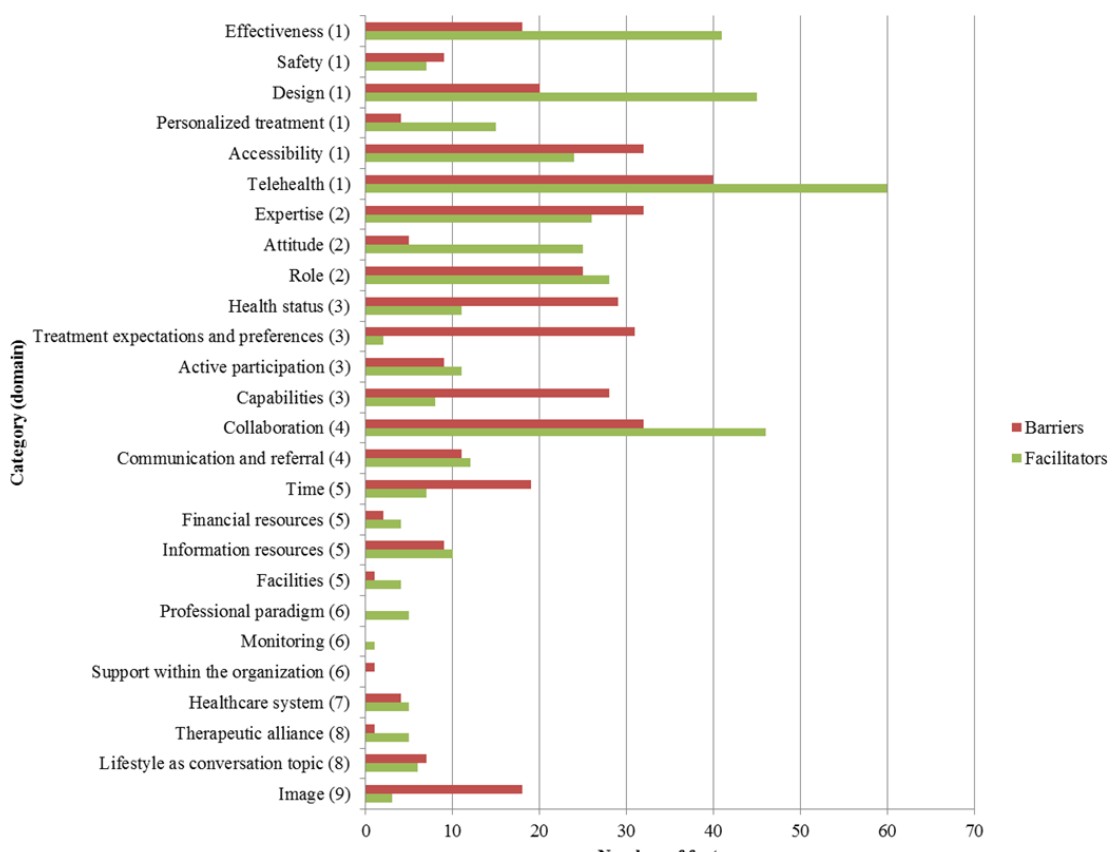

**Figure 2** Overview of the number of barriers and facilitators per category. The domain numbers indicated in brackets refer to the domains as presented in table 3: (1) intervention factors; (2) individual HCP factors; (3) patient factors; (4) professional interactions; (5) incentives and resources; (6) capacity for organisational change; (7) social, political and legal factors; (8) patient and HCP interactions and (9) disease factors. Unclear factors were not included in this figure due to the low number (n=11).

categories: telehealth (n=40), collaboration (n=32), expertise (n=32), accessibility (n=32) and treatment expectations and preferences (n=31). The highest number of facilitators was assigned to the following five categories: telehealth (n=60), collaboration (n=46), design (n=45), effectiveness (n=41) and role (n=28).

**Subcategories**
Tables 4 and 5 present the rankings of the ten largest subcategories of barriers and facilitators respectively. The first place in both rankings was assigned to a subcategory related to interdisciplinary collaboration or healthcare provision.

| Table 4 | Ranking of the ten largest subcategories of barriers | |
|---|---|---|
| **Rank** | **Subcategory of barriers (domain)** | **Factors (n)** |
| 1 | Non-optimal interdisciplinary collaboration or healthcare provision (4—professional interactions) | 31 |
| 2 | Negative attitude towards LIs (3—patient factors) | 28 |
| 3 | Low health literacy (3—patient factors) | 24 |
| | Lack of knowledge or skills around LIs or promoting behavioural change (2—individual HCP factors) | 24 |
| 5 | Perception of own role potentially impeding prescription or follow-up of LIs (2—individual HCP factors) | 23 |
| 6 | Severity of disease and symptoms (3—patient factors) | 17 |
| 7 | Other challenges for HCPs regarding feasibility of telehealth (1—intervention factors) | 16 |
| 8 | LIs have little or no effect on OA (1—intervention factors) | 14 |
| 9 | Lack of time within patient consultations (5—incentives and resources) | 12 |
| | LIs are unavailable or inaccessible (1—intervention factors) | 12 |

HCP, healthcare professional; LI, lifestyle intervention; OA, osteoarthritis.

| Table 5 | Ranking of the ten largest subcategories of facilitators | |
| --- | --- | --- |
| **Rank** | **Subcategory of facilitators (domain)** | **Factors (n)** |
| 1 | Good interdisciplinary collaboration or healthcare provision, or suggestions for improvement (4—professional interactions) | 40 |
| 2 | Perception of own role potentially stimulating prescription or follow-up of LIs (2—individual HCP factors) | 27 |
| 3 | Positive experiences with or suggestions to improve the content or structure of LIs (1—intervention factors) | 24 |
| 4 | Positive attitude towards LIs (2—individual HCP factors) | 22 |
| 5 | Positive attitude or needs of HCPs regarding feasibility of telehealth (1—intervention factors) | 18 |
| | Ease for HCPs during delivery of LIs (1—intervention factors) | 18 |
| 7 | LIs have positive effects on affected joint(s) (1—intervention factors) | 17 |
| 8 | Patient-related benefits regarding feasibility of telehealth (1—intervention factors) | 16 |
| 9 | Ability and importance of providing personalised treatment within LIs (1—intervention factors) | 15 |
| 10 | Having or improving knowledge or skills around LIs or promoting behavioural change (2—individual HCP factors) | 14 |

HCP, healthcare professional; LI, lifestyle intervention.

## DISCUSSION

The aim of this review was to provide an overview of barriers and facilitators that primary and secondary HCPs perceive for implementing LIs in patients with hip and/or knee OA. By linking the identified factors to a framework that was largely based on the TICD checklist,[17] a comprehensive overview of influencing factors was created that could serve as a basis for improving the implementation of LIs within primary and secondary OA care. The variety of domains shows that multiple levels (ie, both the level of the individual HCP and several environmental levels) should be considered in order to achieve this. Within this framework, the extracted barriers were most frequently related to non-optimal interdisciplinary collaboration, a negative attitude of patients towards LIs, low health literacy of patients, and a lack of knowledge and skills of HCPs around LIs or promoting behavioural change. The extracted facilitators were most frequently related to good interdisciplinary collaboration, a positive perception of HCPs' own role in implementing LIs, the content or structure of LIs, and a positive attitude of HCPs towards LIs.

A relatively large number of studies were included, a majority of which was published in recent years. From these 36 studies, a total of 809 influencing factors were extracted. Although all nine domains were covered, the total number of factors identified within each domain differed greatly, ranging from 7 (capacity for organisational change) to 315 (intervention factors). In addition, a large variation was found in the number of barriers and facilitators between the various categories and subcategories. However, we do not know yet whether the established factor frequency is directly related to the importance of the domain, category or subcategory in question. So the fact that we found the highest number of factors within certain domains, categories or subcategories does not

necessarily mean that these are the most important or relevant in the context of implementation. It could also be an indication that studies to date have mainly focused on these aspects, and that the others are still underexposed in the available literature. Therefore, we recommend to take all domains into account in future research in order to avoid missing factors that might be highly relevant for the implementation of LIs. The quality assessment of the included studies showed many unknown ratings due to a lack of information about, for example, the applied methods and their rationale. This finding does not have to mean that the studies are of low quality, but it does emphasise the importance of accurate and complete reporting of research using design-specific reporting guidelines.

Our results reflect those of a previous systematic review conducted by Egerton et al,[15] in which the authors synthesised qualitative evidence only on primary care clinicians' views on providing recommended management of OA up to August 2016. In addition to exercise and weight loss, recommended management included education, self-management support, and medication. The authors identified four barriers as main themes (1): 'OA is not that serious'; (2) 'clinicians are, or perceive they are, underprepared'; (3) 'personal beliefs at odds with providing recommended practice' and (4) 'dissonant patient expectations'. A few system-related factors (eg, time, payment system) were mentioned, but these were not found to be themes across multiple studies. The added value of the current review in comparison to the review by Egerton et al is that factors related to interdisciplinary collaboration and the organisational and societal context were in fact identified. Although these domains were relatively small in terms of number of factors, the current review shows that these factors can also influence the implementation of LIs and thus offers an even broader perspective on the

implementation status of LIs within OA care. Besides an expansion of the review's scope (ie, the inclusion of quantitative data and the perspectives of secondary HCPs), this broader perspective of our review most likely arises from the date of the search. The vast majority (72%) of the included articles were in fact published in the past 5 years (after Egerton *et al* had conducted their review), which shows that there is growing attention for the role of lifestyle as treatment for hip and/or knee OA. Very recently another scoping review has been published, conducted by Nissen *et al*,[63] which focused on clinicians' beliefs and attitudes about physical activity and exercise therapy as treatment for hip and/or knee OA. The authors thematically analysed qualitative data from four types of HCPs (physiotherapists, general practitioners, orthopaedic surgeons and rheumatologists). Their main finding is that many clinicians perceive OA to be a low priority 'wear and tear' disease. In addition, they identified a relative lack of knowledge about and interest in physical activity and exercise management among many clinicians. These findings are also reflected in our results (especially in the domains disease factors and individual HCP factors). In addition, even more barriers and facilitators have been identified in the current review. Compared with this review by Nissen *et al*, our review again has a broader scope (ie, the inclusion of weight management, quantitative data and the perspectives of more types of HCPs) and can therefore be seen as relevant addition to the existing literature on this topic.

In addition to summarising the existing literature on barriers and facilitators for implementing LIs, this review aimed to identify potential gaps in literature on the participation of HCPs. Although we aimed to include perceptions of various primary and secondary HCPs, the results show that studies to date have mainly focused on the views of physiotherapists and general practitioners. These primary HCPs may well be the first point of contact for patients within the care pathway, yet we recommend that other relevant disciplines—like dieticians, lifestyle counsellors, practice nurses and orthopaedic clinicians—be more involved in follow-up research, allowing for a more complete understanding of the patient journey in OA care. Special attention should then be drawn to potential differences in perceived barriers and facilitators between types of HCPs, so that implementation strategies can be tailored as much as possible to the various types of HCPs and their clinical practice.

The resulting overview of barriers and facilitators can be used to improve the implementation of LIs in daily practice. This overview presents factors that are relevant for individual HCPs, as well as for policy-makers, who can facilitate the organisational and societal context in which primary and secondary HCPs work. When developing implementation strategies, possible interactions between the various domains should also be considered. For instance, more time (domain 5) can be used in various ways by HCPs: for their own education (domain 2), provision of information to patients (domain 3),

or interdisciplinary consultation (domain 4). Another example is that societal changes in health insurance or payment structures (domain 7) can lead to increased accessibility of LIs (domain 1), and that limited financial resources might be less of an obstacle for patients (domain 3). Hence changes related to the established factors can have positive effects on multiple levels.

Within the domain of intervention factors, a separate category was created for factors specific to delivering LIs via telehealth. Attention for this modality of healthcare provision has been growing for some time.[64] In addition, during the course of the current review the COVID-19 pandemic emerged, which meant that many HCPs actually had to use telehealth in their daily practice.[65] Although telehealth was not a specific focus of this review, it could be interesting to further investigate the experiences with telehealth and its value for long-term counselling of patients with hip and/or knee OA on behavioural change.[66]

To the best of our knowledge, this is the first review to focus specifically on the implementation of LIs as conservative treatment for hip and/or knee OA while taking into account the perceptions of all primary and secondary HCPs involved. Both qualitative and quantitative data were included, providing broad insight into the topic. All included studies were conducted in North America, Europe and Oceania. Given that the majority of these studies were conducted quite recently, our results are expected to be representative of the current situation in these continents.

There are also a few limitations to acknowledge. First, 'implementing LIs' was defined very broadly and can be seen as an umbrella term, ranging from mentioning a healthy lifestyle to running specific lifestyle programmes. Due to the heterogeneity of the included studies in terms of study design and evaluated LIs, no distinction was made between the different ways of implementing LIs during data analysis. Consequently, the identified barriers and facilitators may not fit with every single way of implementing LIs, but may rather provide insight into the full spectrum of influencing factors. Although data synthesis has not been performed separately for physical activity and weight management either, the created overview gives us the overall impression that barriers and facilitators related to these two lifestyle components are quite similar. One barrier that seems to be unique to weight management is the perception of it being a difficult or sensitive subject to discuss. Regarding physical activity, the perception that it is unsafe or has negative effects seems to be a unique barrier. Second, although data extraction and quality assessment were performed by two researchers independently, data analysis was performed primarily by one researcher. By discussing the resulting classification of factors and any doubts during the process with members of the research team, we aimed to increase the reliability of our findings. Third, the chosen cut-off percentage for extracting quantitative data was based on other scoping reviews combining the results of quantitative

and qualitative studies.[24 25] Therefore, there is a chance that factors that would have been extracted when using a lower cut-off percentage are missing. However, it is also possible that these factors were already extracted from the other included studies and therefore still included in our results. Lastly, as we did not search grey literature there is a slight chance that relevant studies may have been missed.

The comprehensive overview of barriers and facilitators for implementing LIs in patients with hip and/or knee OA by HCPs resulting from this review can serve as a basis for further research and the development of implementation strategies that focus on both the individual and the environmental context of HCPs. However, what the relative importance of the identified factors is and whether differences exist between the various types of primary and secondary HCPs with respect to these factors are not known yet. Further research is required to provide more insight into this relative importance and therewith the most relevant targets for change in daily practice.

## CONCLUSION

This review has shown that multiple factors influence whether or not HCPs implement LIs when treating patients with hip and/or knee OA. Data analysis has resulted in a comprehensive overview of influencing factors, where barriers and facilitators have been subdivided into nine domains, both at an individual and at several environmental levels. The review contributes to existing knowledge about the implementation of LIs by identifying multiple factors related to the intervention, interdisciplinary collaboration and the organisational and societal context. The broad inventory created in this review can be a first step towards an improved implementation of LIs by HCPs in OA care. Future research in this area should focus on determining the relative importance of the identified factors involving all relevant disciplines of primary and secondary HCPs.

**Author affiliations**
[1]Department of Orthopedics, University of Groningen, University Medical Center Groningen, Groningen, The Netherlands
[2]Center for Human Movement Sciences, University of Groningen, University Medical Center Groningen, Groningen, The Netherlands
[3]Department of Rehabilitation Medicine, University of Groningen, University Medical Center Groningen, Groningen, The Netherlands
[4]Peter Harrison Centre for Disability Sport, School of Sport, Exercise and Health Sciences, Loughborough University, Loughborough, UK

**Correction notice** This article has been corrected since it was first published. Table 2 and 3 has been updated.

**Acknowledgements** The authors would like to thank Truus van Ittersum for her advice on search strategies and Annemarie Jenks for her contribution to the study selection process.

**Contributors** Conception and design: SB, RD, LHVvdW, MS and IvdA-S. Collection and assembly of data: SB and JvB. Analysis and interpretation of the data: SB, MS and IvdA-S. Drafting of the article: SB. Critical revision of the article for important intellectual content: SB, JvB, RD, LHVvdW, MS and IvdA-S. Final approval of the article: SB, JvB, RD, LHVvdW, MS and IvdA-S. Obtaining of funding: SB, RD, LHVvdW, MS and IvdA-S. Guarantor: SB.

**Funding** This study is supported by a scholarship from the University of Groningen/University Medical Center Groningen (grant/award number: not applicable).

**Competing interests** None declared.

**Patient consent for publication** Not applicable.

**Ethics approval** This study does not involve human participants.

**Provenance and peer review** Not commissioned; externally peer reviewed.

**Data availability statement** Data sharing not applicable as no datasets generated and/or analysed for this study.

**ORCID iDs**
Sjoukje E Bouma http://orcid.org/0000-0002-8056-6586
Ron L Diercks http://orcid.org/0000-0001-9873-208X
Lucas H V van der Woude http://orcid.org/0000-0002-8472-334X
Martin Stevens http://orcid.org/0000-0001-8183-6894
Inge van den Akker-Scheek http://orcid.org/0000-0002-1614-8419

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
