## [Reviewer comments · BMJ Open]

ARTICLE DETAILS

TITLE (PROVISIONAL)	Barriers and facilitators perceived by healthcare professionals for implementing lifestyle interventions in patients with osteoarthritis: a scoping review
AUTHORS	Bouma, Sjoukje; van Beek, Juliette; Diercks, Ron; van der Woude, Lucas; Stevens, Martin; van den Akker-Scheek, Inge

VERSION 1 – REVIEW

REVIEWER	Ho, Lai-Ming The University of Hong Kong, School of Public Health
REVIEW RETURNED	27-Sep-2021

GENERAL COMMENTS	This is a systematic scoping review, aiming to identify the barriers and facilitators perceived by primary and secondary healthcare professionals (mostly from general practitioners and physiotherapists) for the implementation of lifestyle interventions (physical activity and/or weight management as key components) in patients with hip/knee osteoarthritis. Combining both the results from quantitative and qualitative studies can be challenging. It was written that “factors were only extracted if $\geq 50\%$ of participants indicated that the factor influenced the implementation of LIs” (Line 167). I understand that some studies (refs 24 and 25) adopted this approach, but it seems strange. For example, the qualitative results from Davis (ref 29) study were just based on 3 physiotherapists, whereas quantitative results from Holden (ref 60) study were based on 538 physiotherapists. A factor identified in Holden study can be more important than the one identified in Davis study, even 1% of physiotherapists indicated that the factor influenced the implementation of LIs. This may probably a limitation of interpreting results from both quantitative and qualitative studies. The Methods described in this manuscript are clear and detailed. The results on the perceived barriers and facilitators were presented in terms of nine domains. However these barriers and facilitators may be perceived differently by different healthcare professionals. For example, Holden (ref 60) study among physiotherapists was identified to have 14 out of 42 (ie 33%) factors in patient domain, whereas Cottrell (ref 52) study among general practitioners was found to have 4 out of 31 (ie 13%) factors in the same domain. So a factor considered to be important by one type of health professionals may not be the same by other types of health professionals. Most studies include general practitioners and physiotherapists as study population (Line 223), it may be useful, if possible, to report and discuss the findings by types of healthcare professionals. If the results can be presented
---

	separately, the results will be more useful for further research direction, and the strategies on overcoming the barriers can be more specific and targeted.
--	--

REVIEWER	Taylor, Nicholas La Trobe University, College of Science Health and Engineering
REVIEW RETURNED	01-Nov-2021

GENERAL COMMENTS	This systematic scoping review extracted data on healthcare professionals' perceptions of implementing lifestyle interventions for people with hip or knee osteoarthritis. From a yield of 36 qualitative and survey studies the authors concluded that multiple factors influence implementation of lifestyle factors. The review has been prepared meticulously with attention to detail and process. The review was registered prospectively, a comprehensive search strategy has been completed, a clear and appropriate method of methodological quality applied, and the barriers and facilitators have been linked or mapped to 9 domains based on the Tailored Implementation for Chronic Disease (TICD) Checklist. My main concern is that the synthesis of results that there are 'multiple individual and environmental' factors does not lead to a clear pathway forward in implementing lifestyle interventions in this population. Part of the difficulty may be that the use of the TICD Checklist with its 7 domains plus the 2 further domains makes it difficult to see the wood for the trees. I understand this has been touched on in one paragraph in the discussion but I am not sure it is sufficient. The authors have stated, appropriately I think, that they did not apply a vote-counting content type analysis. I wondered if some further thematic analysis across the TICD checklist might result in more definitive findings on clinician perceptions. A second lesser concern is that some further discussion be included on how the review adds to that of Egerton et al., who incidentally took a quite different analytic approach. Given there appeared to be few studies including the perceptions on secondary HCFs, the review might be strengthened by emphasising how the results complement those of the previous review (e.g. including survey data, greater body of literature, and perhaps the conclusions). Related to this comment, could the first dot point under strengths and limitations be amended? Some specific comments Methods (line 123): Please define what is meant by primary and secondary HCPs. Methods (study selection): I suggest setting up that study type included qualitative and quantitative survey data on HCP perceptions.
--

REVIEWER	Moseng, Tuva Diakonhjemmet Sykehus AS
REVIEW RETURNED	06-Nov-2021

GENERAL COMMENTS	Thank you for the opportunity to review this manuscript on barriers and facilitators perceived by healthcare professionals for implementing lifestyle interventions in patients with OA. The question under study is important to investigate to allow for more successful implementation of recommended core treatments for people with OA in the future. I do, however, have some comments regarding the manuscript.
---

	Introduction  - Please define from the beginning which OA joints you are referring to, as the strength of the treatment recommendations regarding conservative treatments vary between OA sites. - The terms physical activity and exercise are mixed without further explanation. Please specify and define what you are referring to. - I do not recommend to use the term lifestyle interventions in this setting as exercise and weight loss are regarded specific treatments which should be individually tailored to successfully affect OA pain and disability - The review would be more useful and to the point if the term “lifestyle interventions”, which is a very broad and poorly defined term could be replaced with e.g. the recommended first-line / core treatments for hip and knee OA: patient education, exercise and weight loss/control. Or potentially physical activity and weight loss. Such specification of the terms would make the paper much more to-the-point and relevant for both healthcare professionals and researchers in the area. - Please be more specific when using the term “implementation” How is this term defined? Methods  - Please define the term scoping review to improve understanding of why this was the most appropriate method for this review Results  - The results are well presented, although the reported number of barriers and facilitators is large. Would it be possible to design a summarizing figure considering the domains and the specific barriers and facilitators to improve the readability of the results section and provide an easy overview of the results?
--	--

REVIEWER	Tan, Bryan National Healthcare Group Woodlands Health Campus, Orthopaedic Surgery
REVIEW RETURNED	12-Nov-2021

GENERAL COMMENTS	Thank you for inviting me to review this manuscript. Overall impression This study addresses a gap in the literature that the authors have identified Overall, methodologically robust using an established scoping review methodology, systematic search strategy, evaluation of quality through the MMAT, registration on PROSPERO, reporting through the PRISMA-Scr guidelines Analysis guided by the established TICD checklist Overall applicability – the overall broad nature of this scoping review does provide a broad sense of the multiple of factors relevant to osteoarthritis that can be grouped broadly according to TICD framework that healthcare professionals or policymakers can use. It might have been helpful to have a deeper analysis of the data rather than a general listing of all the factors grouped fairly strictly according to the TICD framework to (1) identify specific key or new
---

	insights/priorities that HCP or policymakers can perhaps focus on and (2) to identify other gaps in the literature for future research which the scoping review is ideally designed to achieve. Specific Comments Line 52 – manuscript length too long Line 87 – definition of primary and secondary healthcare professional? Different healthcare system use different nomenclature so it would be good to be clear here as the barriers between both can be significantly different. Line 114 and 119 – any start date for the search? Would very old publications > 20 years ago may be less relevant in today's context? Line 129-136 – were psychosocial intervention considered as part of the definition? There is a fair amount of overlap between lifestyle interventions and psychosocial interventions and it would be good to be clear regarding this. Lifestyle intervention is a very broad term in itself and can encompass simple lifestyle advice (exercise, weight loss) to a comprehensive program lead by healthcare professional. Again, the barriers for both will be quite different and it will be important to distinguish/separate as part of the analysis rather than a simple listing of factors. Line 197-199 – it looks like a purely deductive approach was used where factors were fitted into the TICD framework and checklist. Was an inductive approach used? What about the factors that did not fit into the TICD framework? Line 194-197 – authors indicated that 2 additional domains were added to the TICD framework based on their own unpublished research. Given that this data is not published yet for readers to reference, could the authors provide a bit of background to understand how these 2 additional domains were added. Noted that the TICD checklist is actually a work in progress and has been used in various contexts and studies. Line 217-221 – noted all the studies were from a western population and were predominately qualitative in nature Line 230 – overall general quality of studies are quite poor based on the MMAT although the authors did comment in the discussion that this may not be necessarily indicative of poor quality but rather poor reporting in accordance to the guidelines. Line 246 – it would be good to provide a descriptive diagram for pictorial appreciation of the spread of the data and ease of understanding Line 379-386 - In terms of gap identified, it only highlighted the need for research to be done in barriers to implement LI by secondary HCP for OA. Were there any other gaps identified for further research? Were any of the domains highlighted noted to have potential for future research?
--	---

VERSION 1 – AUTHOR RESPONSE

Reactions to the comments of Reviewer 1:
Dr. Lai-Ming Ho, The University of Hong Kong

This is a systematic scoping review, aiming to identify the barriers and facilitators perceived by primary and secondary healthcare professionals (mostly from general practitioners and physiotherapists) for the implementation of lifestyle interventions (physical activity and/or weight management as key components) in patients with hip/knee osteoarthritis.

- Author response 6: Thank you for reviewing our manuscript and for your comments on the data extraction approach and the presentation of results respectively.

Combining both the results from quantitative and qualitative studies can be challenging. It was written that “factors were only extracted if $\geq 50\%$ of participants indicated that the factor influenced the implementation of LIs” (Line 167). I understand that some studies (refs 24 and 25) adopted this approach, but it seems strange. For example, the qualitative results from Davis (ref 29) study were just based on 3 physiotherapists, whereas quantitative results from Holden (ref 60) study were based on 538 physiotherapists. A factor identified in Holden study can be more important than the one identified in Davis study, even 1% of physiotherapists indicated that the factor influenced the implementation of LIs. This may probably a limitation of interpreting results from both quantitative and qualitative studies.

- Author response 7: We understand your doubts about this cut-off percentage. Although we based our approach on previous scoping reviews combining the results of quantitative and qualitative studies [references 24 and 25], this cut-off percentage can be considered an arbitrary value. By using this cut-off percentage, we excluded factors that may have a significant influence on the implementation of LIs. However, given the relatively large number of included studies, it might also be the case that these “missing” factors have been extracted from the other studies and were therefore still captured in our results. We followed your suggestion and added a short reflection on our data extraction approach in the limitation section of the discussion.
 - MC Line 510-515: Third, the chosen cut-off percentage for extracting quantitative data was based on other scoping reviews combining the results of quantitative and qualitative studies^{24,25}. Therefore, there is a chance that factors that would have been extracted when using a lower cut-off percentage are missing. However, it is also possible that these factors were already extracted from the other included studies and therefore still included in our results.

The Methods described in this manuscript are clear and detailed. The results on the perceived barriers and facilitators were presented in terms of nine domains. However these barriers and facilitators may be perceived differently by different healthcare professionals. For example, Holden (ref 60) study among physiotherapists was identified to have 14 out of 42 (ie 33%) factors in patient domain, whereas Cottrell (ref 52) study among general practitioners was found to have 4 out of 31 (ie 13%) factors in the same domain. So a factor considered to be important by one type of health professionals may not be the same by other types of health professionals. Most studies include general practitioners and physiotherapists as study population (Line 223), it may be useful, if possible, to report and discuss the findings by types of healthcare professionals. If the results can be presented separately, the results will be more useful for further research direction, and the strategies on overcoming the barriers can be more specific and targeted.

- Author response 8: We agree that it would be very useful to know whether the perceived barriers and facilitators are different for the various types of HCPs. However, comparing the

results between the different types of HCPs was beyond the scope of this review. When looking at our data again, it also appears to be very complex to present the results separately by types of HCPs because of two main reasons. First, it was not possible for all factors to determine by which HCP the factor was reported (particularly in the case of focus group data). Second, due to the heterogeneity of study designs and evaluated ways of implementing LIs, it would be difficult to draw conclusions from the data. Therefore, we think that separating the results by types of HCPs would be at the expense of the interpretability of the findings in our review. We certainly share your impression that there might be relevant differences in the perceptions of different HCPs, and have therefore added this as a point of interest for future research.

- MC Line 463-465: Special attention should then be drawn to potential differences in perceived barriers and facilitators between types of HCPs, so that implementation strategies can be tailored as much as possible to the various types of HCPs and their clinical practice.

Reactions to the comments of Reviewer 2:

Prof. Nicholas Taylor, La Trobe University, Eastern Health

This systematic scoping review extracted data on healthcare professionals' perceptions of implementing lifestyle interventions for people with hip or knee osteoarthritis. From a yield of 36 qualitative and survey studies the authors concluded that multiple factors influence implementation of lifestyle factors. The review has been prepared meticulously with attention to detail and process. The review was registered prospectively, a comprehensive search strategy has been completed, a clear and appropriate method of methodological quality applied, and the barriers and facilitators have been linked or mapped to 9 domains based on the Tailored Implementation for Chronic Disease (TICD) Checklist.

- Author response 9: Thank you for reviewing our manuscript and for your positive feedback on the preparation of our review.

My main concern is that the synthesis of results that there are 'multiple individual and environmental' factors does not lead to a clear pathway forward in implementing lifestyle interventions in this population. Part of the difficulty may be that the use of the TICD Checklist with its 7 domains plus the 2 further domains makes it difficult to see the wood for the trees. I understand this has been touched on in one paragraph in the discussion but I am not sure it is sufficient. The authors have stated, appropriately I think, that they did not apply a vote-counting content type analysis. I wondered if some further thematic analysis across the TICD checklist might result in more definitive findings on clinician perceptions.

- Author response 10: We agree that the way in which we presented our results did not lead to concrete recommendations for improving the implementation of lifestyle interventions. In the initial version of the manuscript, we were hesitant to quantify the barriers and facilitators found. As we do not know yet whether the frequency of extraction is related to the actual importance of the factor regarding implementation of lifestyle interventions in daily practice, we did not want to emphasize these factor frequencies too much. However, comments from you and the other reviewers have made us reconsider this approach for synthesis of results. Therefore, in this revised version of the manuscript, we have chosen to not only quantify the domains, but also the categories and subcategories of barriers and facilitators, in order to show more concrete recommendations for future research and daily practice. As a result we rewrote part of the results section. In addition, we placed more emphasis in the discussion on the unknown relationship between factor frequency and relevance in order to avoid misinterpretation of our findings.
 - Figure 2, Table 4 and Table 5
 - MC Page 12-16: paragraph "Synthesis of results"

- MC Line 395-401: Within this framework, the extracted barriers were most frequently related to non-optimal interdisciplinary collaboration, a negative attitude of patients toward LIs, low health literacy of patients, and a lack of knowledge and skills of HCPs around LIs or promoting behavioral change. The extracted facilitators were most frequently related to good interdisciplinary collaboration, a positive perception of HCPs' own role in implementing LIs, the content or structure of LIs, and a positive attitude of HCPs toward LIs.
- MC Line 407-417: In addition, a large variation was found in the number of barriers and facilitators between the various categories and subcategories. However, we do not know yet whether the established factor frequency is directly related to the importance of the domain, category or subcategory in question. So the fact that we found the highest number of factors within certain domains, categories or subcategories does not necessarily mean that these are the most important or relevant in the context of implementation. It could also be an indication that studies to date have mainly focused on these aspects, and that the others are still underexposed in the available literature. Therefore, we recommend to take all domains into account in future research in order to avoid missing factors that might be highly relevant for the implementation of LIs.

A second, lesser concern is that some further discussion be included on how the review adds to that of Egerton et al., who incidentally took a quite different analytic approach. Given there appeared to be few studies including the perceptions on secondary HCFs, the review might be strengthened by emphasising how the results complement those of the previous review (e.g. including survey data, greater body of literature, and perhaps the conclusions). Related to this comment, could the first dot point under strengths and limitations be amended?

- Author response 11: We expanded the section in the discussion in which we compare our findings with previous literature. Regarding the review of Egerton et al., we stated that the additional value of our review can be found in the combination of broader inclusion criteria and a more recently performed search. We also added a short reflection on the very recently published scoping review of Nissen et al. [reference 63]. We think that the added value of our review compared to these two already conducted reviews is now presented more clearly in this section.
- MC Line 431-453: The added value of the current review in comparison to the review by Egerton et al. is that factors related to interdisciplinary collaboration and the organizational and societal context were in fact identified. Although these domains were relatively small in terms of number of factors, the current review shows that these factors can also influence the implementation of LIs and thus offers an even broader perspective on the implementation status of LIs within OA care. Besides an expansion of the review's scope (i.e. the inclusion of quantitative data and the perspectives of secondary HCPs), this broader perspective of our review most likely arises from the date of the search. The vast majority (72%) of the included articles were in fact published in the past five years (after Egerton et al. had conducted their review), which shows that there is growing attention for the role of lifestyle as treatment for hip and/or knee OA. Very recently another scoping review has been published, conducted by Nissen et al.⁶³, which focused on clinicians' beliefs and attitudes about physical activity and exercise therapy as treatment for hip and/or knee OA. The authors thematically analyzed qualitative data from four types of HCPs (physiotherapists, general practitioners, orthopedic surgeons and rheumatologists). Their main finding is that many clinicians perceive OA to be a low priority "wear and tear" disease. In addition, they identified a relative lack of knowledge about and interest in physical activity and exercise management among many clinicians. These findings are also reflected in our results (especially in the domains disease

factors and individual HCP factors). In addition, even more barriers and facilitators have been identified in the current review. Compared to this review by Nissen et al, our review again has a broader scope (i.e. the inclusion of weight management, quantitative data and the perspectives of more types of HCPs) and can therefore be seen as relevant addition to the existing literature on this topic.

- Additional reference

63. Nissen N, Holm PM, Bricca A, Dideriksen M, Tang LH, Skou ST. Clinicians' beliefs and attitudes to physical activity and exercise therapy as treatment for knee and/or hip osteoarthritis: a scoping review. *Osteoarthritis Cartilage* (online ahead of print). DOI:10.1016/j.joca.2021.11.008

- We revised the first two dot points under “Strengths and limitations of this study” to put more emphasis on the combination of quantitative and qualitative data instead of the inclusion of both primary and secondary HCPs.
 - Strengths and limitations of this study: MC Line 62-69

Some specific comments

*Methods (line 123): Please define what is meant by primary and secondary HCPs.

- Author response 12: The terms “primary HCPs” and “secondary HCPs” refer to both HCPs providing general medical care as well as HCPs providing more specialized care. The accessibility of secondary HCPs can vary by country or by healthcare system, i.e. whether patients need a referral from a primary HCP or not. We added this definition to the method section to describe the study population more clearly.
 - MC Line 143-145: This definition includes, respectively, HCPs providing general medical care and HCPs providing more specialized care (with or without a referral).

*Methods (study selection): I suggest setting up that study type included qualitative and quantitative survey data on HCP perceptions.

- Author response 13: We followed your suggestion and added that barriers and facilitators were based on both quantitative and qualitative data. These study types are also mentioned in the paragraph “Data sources and searches”.
 - MC Line 150-151: These barriers and facilitators were extracted from both quantitative (e.g. survey) and qualitative (e.g. interview) data.

Reactions to the comments of Reviewer 3:

Dr. Tuva Moseng, Diakonhjemmet Sykehus AS

Thank you for the opportunity to review this manuscript on barriers and facilitators perceived by healthcare professionals for implementing lifestyle interventions in patients with OA. The question under study is important to investigate to allow for more successful implementation of recommended core treatments for people with OA in the future. I do, however, have some comments regarding the manuscript.

- Author response 14: Thank you for reviewing our manuscript. We appreciate your positive feedback on the potential contribution of our study and your suggestions to improve the quality of our manuscript.

Introduction

- Please define from the beginning which OA joints you are referring to, as the strength of the treatment recommendations regarding conservative treatments vary between OA sites.

- Author response 15: Thank you for pointing this out. We added “hip and/or knee” to various sentences in the introduction section to clarify which joints we are referring to.
 - MC Line 78, 80-81, 84 and 94: hip and/or knee OA

- The terms physical activity and exercise are mixed without further explanation. Please specify and define what you are referring to.

- Author response 16: Indeed both terms are used in the manuscript. The focus of our review was physical activity in the broad sense, ranging from physical activity during activities of daily living to participation in supervised or non-supervised exercise therapy or sports. We only used the term “exercise” when referring to publications in which this term was used. We added the above definition of physical activity to the method section to emphasize the scope of this concept and to clarify that exercise is part of it.
 - MC Line 158-160: Physical activity was also broadly defined, ranging from physical activity during activities of daily living to participation in supervised or non-supervised exercise therapy or sports.

- I do not recommend to use the term lifestyle interventions in this setting as exercise and weight loss are regarded specific treatments which should be individually tailored to successfully affect OA pain and disability

- Author response 17: We agree that physical activity and weight loss should be individually tailored to successfully treat hip and/or knee OA. In this review, we grouped both lifestyle aspects under the term “lifestyle interventions” and we did not separate the analysis and synthesis of data. Due to the heterogeneity of the included studies, this would be a complex comparison (if possible at all) that probably would not benefit the interpretability of our results. We acknowledge that physical activity and weight loss are distinct treatments, and therefore that different barriers and facilitators could be perceived in relation to these two lifestyle aspects. Although data analysis was not formally performed separately, we had already included a comment about the distinction between physical activity and weight loss in the limitation section of the discussion. Here we indicated that our impression is that, apart from two unique barriers, there do not appear to be major differences between the influencing factors regarding both lifestyle aspects.
 - MC Line 501-506: Although data synthesis has not been performed separately for physical activity and weight management either, the created overview gives us the overall impression that barriers and facilitators related to these two lifestyle components are quite similar. One barrier that seems to be unique to weight management is the perception of it being a difficult or sensitive subject to discuss. Regarding physical activity, the perception that it is unsafe or has negative effects seems to be a unique barrier.

- The review would be more useful and to the point if the term “lifestyle interventions”, which is a very broad and poorly defined term could be replaced with e.g. the recommended first-line / core treatments for hip and knee OA: patient education, exercise and weight loss/control. Or potentially physical activity and weight loss. Such specification of the terms would make the paper much more to-the-point and relevant for both healthcare professionals and researchers in the area.

- Author response 18: Thank you for this suggestion. We agree that lifestyle interventions in general is a very broad term and covers more lifestyle aspects than just physical activity and weight loss (e.g. smoking, stress and sleep). We took your suggestion into consideration and further discussed the terminology we used. However, since we clearly defined in the method section that physical activity and weight management were the focus of this review and to stay in line with our previous publication on this topic [reference 26], we nevertheless decided not to change the terminology.
 - MC Line 151-154: Implementing LIs was broadly defined, ranging from mentioning or discussing a healthy lifestyle to recommending or running specific lifestyle programs, as long as it was clearly described that physical activity and/or weight management were key components.

- Please be more specific when using the term “implementation” How is this term defined?
 - Author response 19: In the context of our review, implementation refers to the use of LIs as conservative treatment for hip and/or knee OA by individual HCPs. The ways in which HCPs can do this have been described in the method section (“ranging from mentioning or discussing a healthy lifestyle to recommending or running specific lifestyle programs”). We added the above definition of implementation to the introduction to clarify the scope of our review.
 - MC Line 107-109: Within the context of this review, implementation was defined as the use of LIs as conservative treatment for hip and/or knee OA by individual HCPs.

Methods

- Please define the term scoping review to improve understanding of why this was the most appropriate method for this review
 - Author response 20: We added the definition of a scoping review from Colquhoun et al. to this paragraph. We think this definition explains why the methodology of a scoping review was chosen to answer our research question.
 - MC Line 114-120: A scoping review has been defined as follows by Colquhoun et al.: “a form of knowledge synthesis that addresses an exploratory research question aimed at mapping key concepts, types of evidence and gaps in research related to a defined area or field by systematically searching, selecting, and synthesizing existing knowledge”¹⁸. Therefore, a scoping review was considered a suitable methodology to summarize existing literature on barriers and facilitators for implementing LIs in hip and/or knee OA and to identify potential gaps in the current literature on participation of primary and secondary HCPs.

Results

- The results are well presented, although the reported number of barriers and facilitators is large. Would it be possible to design a summarizing figure considering the domains and the specific barriers and facilitators to improve the readability of the results section and provide an easy overview of the results?
 - Author response 21: Thank you for this suggestion. We added Figure 2, which shows the number of barriers and facilitators per category. We also created two extra tables (Table 4 and Table 5), in which the top 10 subcategories of barriers and the top 10 subcategories of facilitators are presented respectively. In addition, we rewrote part of the results section. We think these changes will improve the readability and interpretability of our findings.
 - Figure 2, Table 4 and Table 5
 - MC Page 12-16: paragraph “Synthesis of results”

Reactions to the comments of Reviewer 4:

Dr. Bryan Tan, National Healthcare Group Woodlands Health Campus

Thank you for inviting me to review this manuscript.

Overall impression

This study addresses a gap in the literature that the authors have identified

Overall, methodologically robust using an established scoping review methodology, systematic search strategy, evaluation of quality through the MMAT, registration on PROSPERO, reporting through the PRISMA-Scr guidelines

Analysis guided by the established TICD checklist

Overall applicability – the overall broad nature of this scoping review does provide a broad sense of the multiple of factors relevant to osteoarthritis that can be grouped broadly according to TICD framework that healthcare professionals or policymakers can use.

It might have been helpful to have a deeper analysis of the data rather than a general listing of all the factors grouped fairly strictly according to the TICD framework to (1) identify specific key or new insights/priorities that HCP or policymakers can perhaps focus on and (2) to identify other gaps in the literature for future research which the scoping review is ideally designed to achieve.

- Author response 22: Thank you for reviewing our manuscript and for your positive feedback on the applied methodology and the potential implications of our study. We agree that a deeper analysis of the data could be helpful to identify priorities and/or other literature gaps. We will further elaborate on this in response to your specific comments below.

Specific Comments

Line 52 – manuscript length too long

- Author response 23: By mistake, it was stated in the initial version that the maximum word count was 4000, while this was actually a recommended number of words. After revising our manuscript, the word count is 4152, which is slightly less than the word count of the initial version (4176).

Line 87 – definition of primary and secondary healthcare professional? Different healthcare system use different nomenclature so it would be good to be clear here as the barriers between both can be significantly different.

- Author response 24: The term primary and secondary HCPs was defined as both HCPs providing general medical care as well as HCPs providing more specialized care. We realize that the accessibility of secondary care can indeed differ per country or per healthcare system. In some cases, patients need a referral from a primary HCP to visit a secondary HCP. In other cases, patients can consult a secondary HCP by self-referral. We added our definition of primary and secondary HCPs to the method section to clarify that we mean both general medical care and more specialist care, independent of the national healthcare system organization.
 - MC Line 143-145: This definition includes, respectively, HCPs providing general medical care and HCPs providing more specialized care (with or without a referral).

Line 114 and 119 – any start date for the search? Would very old publications > 20 years ago may be less relevant in today's context?

- Author response 25: We did not set a specific start date for the search, and added “from inception” to clarify this. We agree that publications over 20 years old may be less relevant to current clinical practice. However, the oldest article included in our review was published in 2006, and the vast majority (72%) of the included studies were conducted in the past five years.
 - MC Line 132-133: to identify relevant articles from inception up to 19 January 2021

Line 129-136 – were psychosocial intervention considered as part of the definition? There is a fair amount of overlap between lifestyle interventions and psychosocial interventions and it would be good to be clear regarding this. Lifestyle intervention is a very broad term in itself and can encompass simple lifestyle advice (exercise, weight loss) to a comprehensive program lead by healthcare professional. Again, the barriers for both will be quite different and it will be important to distinguish/separate as part of the analysis rather than a simple listing of factors.

- Author response 26: Lifestyle interventions is indeed a very broad term. In our review, we defined the term implementing LIs as follows in the method section: “ranging from mentioning or discussing a healthy lifestyle to recommending or running specific lifestyle programs, as

long as it was clearly described that physical activity and/or weight management were key components". Several of the included articles reported the experiences of HCPs with delivering more comprehensive programs that also included, for example, health coaching or education or cognitive behavioral therapy. We agree that the barriers and facilitators for such a comprehensive lifestyle program might be different compared to only giving lifestyle advice. Moreover, there might also be differences in perceived barriers and facilitators with regard to the same program when comparing HCPs delivering the program face-to-face versus digitally. However, due to the heterogeneity of the included studies in terms of study design and evaluated LIs, we think that separating the results would be a very complex analysis that would negatively affect the interpretability of the findings of this review. Therefore, we added a statement about this in the limitation section of the discussion.

- MC Line 497-501: Due to the heterogeneity of the included studies in terms of study design and evaluated LIs, no distinction was made between the different ways of implementing LIs during data analysis. Consequently, the identified barriers and facilitators may not fit with every single way of implementing LIs, but may rather provide insight into the full spectrum of influencing factors.

Line 197-199 – it looks like a purely deductive approach was used where factors were fitted into the TICD framework and checklist. Was an inductive approach used? What about the factors that did not fit into the TICD framework?

- Author response 27: Yes, an inductive approach was also applied during the data analysis process. The first step was indeed a deductive approach, in which all extracted factors were assigned the one of the nine domains based on the TICD checklist and our own previous research. All factors could be linked to one of these domains, so there was no need to develop any additional domains. The second step of the data analysis process was an inductive approach. Different categories and subcategories of factors were developed inductively per domain. We rephrased this sentence to clarify that data analysis was not a purely deductive approach.
 - MC Line 221-223: One researcher (SB) assigned all extracted factors to one of these nine domains and then inductively developed different categories and subcategories of factors per domain.

Line 194-197 – authors indicated that 2 additional domains were added to the TICD framework based on their own unpublished research. Given that this data is not published yet for readers to reference, could the authors provide a bit of background to understand how these 2 additional domains were added. Noted that the TICD checklist is actually a work in progress and has been used in various contexts and studies.

- Author response 28: In the meantime, the focus group study in question has been accepted for publication. We have therefore added the reference to this previous study. In this (forthcoming) publication, it is explained in detail how the TICD checklist was used during the data analysis process and how the two additional domains were developed.
 - Additional reference

26. Bouma SE, van Beek JFE, Alma MA, Diercks RL, van der Woude LHV, van den Akker-Scheek I, et al. What affects the implementation of lifestyle interventions in patients with osteoarthritis? A multidisciplinary focus group study among healthcare professionals. *Disabil Rehabil* (forthcoming). DOI:10.1080/09638288.2021.2011438

Line 217-221 – noted all the studies were from a western population and were predominately qualitative in nature

- Author response 29: Indeed all included studies have been conducted in western countries. This finding has also been mentioned in the discussion.

- MC Line 490-491: All included studies were conducted in North America, Europe and Oceania.

Line 230 – overall general quality of studies are quite poor based on the MMAT although the authors did comment in the discussion that this may not be necessarily indicative of poor quality but rather poor reporting in accordance to the guidelines.

- Author response 30: Indeed many items of the quality assessment could not be rated due to missing information in the included articles. This is also the reason why we explicitly recommend researchers in the discussion to use design-specific reporting guidelines.
 - MC Line 418-421: The quality assessment of the included studies showed many unknown ratings due to a lack of information about, for example, the applied methods and their rationale. This finding does not have to mean that the studies are of low quality, but it does emphasize the importance of accurate and complete reporting of research using design-specific reporting guidelines.

Line 246 – it would be good to provide a descriptive diagram for pictorial appreciation of the spread of the data and ease of understanding

- Author response 31: Thank you for this suggestion. We added Figure 2, which shows the number of barriers and facilitators per category. We think this figure makes it easier to understand and interpret our results. To improve the readability of the results section, we also rewrote part of it and added two extra tables (Table 4 and Table 5).
 - Figure 2, Table 4 and Table 5
 - MC Page 12-16: paragraph “Synthesis of results”

Line 379-386 - In terms of gap identified, it only highlighted the need for research to be done in barriers to implement LI by secondary HCP for OA. Were there any other gaps identified for further research? Were any of the domains highlighted noted to have potential for future research?

- Author response 32: We highlighted the need for further research among all relevant disciplines involved in OA care as it was our explicit aim to identify potential gaps in literature on the participation of HCPs. Besides this already mentioned gap, you could indeed state that there is also a gap in literature regarding the different domains of our framework. Domains 6 to 9 in particular consist of relatively few factors (varying from 7-21 factors per domain) compared to domains 1 to 5 (varying from 56-315 factors per domain). Rather than concluding that these former domains are less important in the context of implementation, it could also be the case that these domains have been understudied to date. We added a recommendation on including these domains in future research in the discussion.
 - MC Line 416-417: Therefore, we recommend to take all domains into account in future research in order to avoid missing factors that might be highly relevant for the implementation of LIs.

VERSION 2 – REVIEW

REVIEWER	Ho, Lai-Ming The University of Hong Kong, School of PUblic Health
REVIEW RETURNED	10-Dec-2021

GENERAL COMMENTS	Thank you for the revised manuscript. They have satisfactorily addressed the comments, and clarified some important points. The manuscript is considerably improved. I don't have any additional comments. Thank you!
---

REVIEWER	Taylor, Nicholas La Trobe University, College of Science Health and Engineering
REVIEW RETURNED	15-Dec-2021

GENERAL COMMENTS	The authors have completed a thorough revision resulting in an improved manuscript. I think the additional level of synthesis, a form of content analysis, makes the results easier to interpret. The authors have also included some thoughtful discussion about how their scoping review adds to the literature.
--

REVIEWER	Moseng, Tuva Diakonhjemmet Sykehus AS
REVIEW RETURNED	26-Dec-2021

GENERAL COMMENTS	No further comments. All my concerns have been addressed in the revision
--

REVIEWER	Tan, Bryan National Healthcare Group Woodlands Health Campus, Orthopaedic Surgery
REVIEW RETURNED	24-Dec-2021

GENERAL COMMENTS	Thank you for allowing me to review the revised version of the manuscript. The authors have made a great effort to address all the concerns raised up by the various reviewers and edited the manuscript, in particular the results section to make it more reader friendly and to facilitate application by healthcare professionals who are involved in the management and care of knee osteoarthritis patients through identifying the key facilitators and barriers to implementation lifestyle interventions.
--